# Regulation of Female Folliculogenesis by Tsp1a in Nile Tilapia (*Oreochromis niloticus*)

**DOI:** 10.3390/ijms21165893

**Published:** 2020-08-16

**Authors:** Mimi Jie, He Ma, Li Zhou, Jiahong Wu, Minghui Li, Xingyong Liu, Deshou Wang

**Affiliations:** 1Key Laboratory of Freshwater Fish Reproduction and Development (Ministry of Education), Key Laboratory of Aquatic Science of Chongqing, School of Life Sciences, Southwest University, Chongqing 400715, China; jiemm813@163.com (M.J.); zhouli4510@163.com (L.Z.); wujiahong419@163.com (J.W.); limh@163.com (M.L.); 2State Key Laboratory of Marine Environmental Science, College of Ocean and Earth Sciences, Xiamen University, Xiamen 361005, China; mahe712@126.com

**Keywords:** Nile tilapia, *thrombospondin1*, CRISPR/Cas9, folliculogenesis

## Abstract

TSP1 was reported to be involved in multiple biological processes including the activation of TGF-β signaling pathways and the regulation of angiogenesis during wound repair and tumor growth, while its role in ovarian folliculogenesis remains to be elucidated. In the present study, Tsp1a was found to be expressed in the oogonia and granulosa cells of phase I to phase IV follicles in the ovaries of Nile tilapia by immunofluorescence. *tsp1a* homozygous mutants were generated by CRISPR/Cas9. Mutation of *tsp1a* resulted in increased oogonia, reduced secondary growth follicles and delayed ovary development. Expression of the cell proliferation marker PCNA was significantly up-regulated in the oogonia of the mutant ovaries. Furthermore, transcriptomic analysis revealed that expressions of DNA replication related genes were significantly up-regulated, while cAMP and MAPK signaling pathway genes which inhibit cell proliferation and promote cell differentiation were significantly down-regulated. In addition, aromatase (Cyp19a1a) expression and serum 17β-estradiol (E2) concentration were significantly decreased in the mutants. These results indicated that lacking *tsp1a* resulted in increased proliferation and inhibited differentiation of oogonia, which in turn, resulted in increased oogonia, reduced secondary growth follicles and decreased E2. Taken together, our results indicated that *tsp1a* was essential for ovarian folliculogenesis in Nile tilapia.

## 1. Introduction

Ovarian folliculogenesis refers to the formation, development and maturation of female gametes, including oogonia proliferation, oocyte growth and maturation, which requires complex and coordinated interaction of numerous autocrine, paracrine, and endocrine factors [1,2,3,4]. Factors within the extracellular matrix (ECM) have also been reported to offer the complex milieu that facilitates the necessary interaction between ovarian follicles and their surroundings to ensure follicles growth and development. Members of the thrombospondin (TSP) family also belong to this group of ECM proteins [5,6,7,8,9,10]. 

Thrombospondin1 (TSP1), a member of *TSP* gene family, is a conserved matricellular glycoproteins [11,12]. *TSP1* is an important gene expressed in granulosa and theca cells of ovaries in mammals. In bovine, TSP1 was reported to be expressed in ovarian granulosa cells and theca cells [13]. In marmoset, TSP1 expressed in granulosa cells of secondary, tertiary, pre-ovulatory, and atretic follicles [14]. In rat, TSP1 showed a similar pattern, with expression in ovarian granulosa cells during the early antral and ovulatory phases [5]. 

Previously, we reported the cloning of two types of *thrombospondin1* (named *tsp1a* and *tsp1b*) from the Nile tilapia and medaka [15]. Phylogenetic analysis of these *tsp1* sequences, together with those available from other vertebrates further demonstrated that two types of *tsp1* exist only in teleost. In situ hybridization analysis showed that tilapia *tsp1a* and *tsp1b* were each expressed in a wide range of tissues examined. In ovaries, *tsp1a* was detected in ovarian granulosa cells, while *tsp1b* was detected in the ovarian theca cells. Further, the expression pattern of *tsp1a* was positively correlated with the serum E2 level during the spawning cycle of XX tilapia, suggesting that *tsp1a* may be involved in the development of tilapia ovaries by affecting serum E2 level in ovarian somatic cells [15]. These results indicated that *tsp1a* may play an important role in folliculogenesis of tilapia.

The distinct roles of *TSP1* have been investigated extensively by *TSP1* knockout in mice. *TSP1*-null mice demonstrated organizing pneumonia [16], delayed wound healing [17], enhanced postinfarction inflammatory response [18], reduced amount of active TGF-β within glomeruli [19], and higher capillarity in the heart and skeletal muscle [20]. Mice lacking *TSP1* were subfertile and exhibited ovarian hypervascularization and altered ovarian morphology [21]. The functional studies of TSP1 were mainly carried out on injured skin, bone, heart, skeletal muscle, kidney and other tissues [22,23,24], while few studies have investigated the role of TSP1 in the ovaries. The function of *tsp1a* in teleost, especially in gonad development, has not been reported. 

The Nile tilapia (*Oreochromis niloticus*) is an important farmed fish with an XX-XY sex- determination system. The availability of genetic all-XX and all-XY fish [25], shorter spawning cycle (14 days), together with the availability of high-quality genome sequences [26], have made it an excellent model for the study of gene expression and function in relation to reproduction and fertility. Particularly, genome editing methods have been established in tilapia by our group [27,28]. In the present study, we generated Tsp1a antibody and examined its cellular localization in gonad and created *tsp1a* mutant line and analyzed the gonad phenotype, germ cell number, gonadal gene expression, serum E2 level in tilapia.

## 2. Results

### 2.1. Cellular Localization of Tsp1a in Tilapia Ovaries

Quantitative reverse transcription polymerase chain reaction (qRT-PCR) analysis revealed that *tsp1a* mRNA was detected in the ovaries from 5 dah (days after hatching) to 180 dah. The expression level in the ovaries gradually increased during the evaluated period, with a remarkably increased level at 180 dah (the vitellogenic stage) (Figure 1A). Tsp1a polyclonal antibody was produced to characterize the cellular localization in tilapia ovaries. Histological classification of oocytes in the tilapia was according to Coward and Bromage (1998). By immunofluorescence (IF) assay, Tsp1a was detected in the oogonia of the ovaries at 30 dah fish (Figure 1B). Further, Tsp1a was found to be expressed in the granulosa cells of phase I to phase IV follicles in the ovaries of 180 dah fish. Similarly, expression of Tsp1a in the granulosa cells was low in the phase I and II follicles, then increased in phase III and IV follicles (Figure 1C–F).

### 2.2. Generation of tsp1a Homozygous Mutants

The gRNA sites containing HpyAV adjacent to protospacer adjacent motif (PAM) were selected in the third exon of *tsp1a*. Genomic DNA extracted from 20 injected embryos was used as a template for PCR amplification and mutation assays. Complete digestion with HpyAV produced two fragments in the control groups, while an intact DNA fragment was observed in embryos injected with both Cas9 mRNA and target gRNA. Representative Sanger sequencing results from the uncleaved bands were listed. In-frame and frame-shift deletions induced at the target site were confirmed by Sanger sequencing (Figure 2A). The *tsp1a* F0 XY mutants were generated by CRISPR/Cas9. F1 mutant fish were obtained by crossing chimeric F0 XY males with WT XX females. Heterozygous F1 offspring with a deletion of 7 base-pairs for *tsp1a* were selected to breed the F2 generation (Figure 2B). Restriction enzyme digestion assay identified the *tsp1a*^+/+^, *tsp1a*^+/−^ and *tsp1a*^−/−^ individuals (Figure 2C). Further, homozygous mutant fish of *tsp1a* were validated by Sanger sequencing (Figure 2D). Immunofluorescence assay confirmed that a specific signal of Tsp1a was detected in the *tsp1a*^+/+^ ovaries while absent in the *tsp1a*^−/−^ ovaries (Figure 2E,F). Consistently, western blot analyses confirmed that a specific band of the Tsp1a protein with the expected size of 128 KDa was detected in the *tsp1a*^+/+^, but not in the *tsp1a*^−/−^ ovaries (Figure 2G)

### 2.3. Homozygous Mutation of tsp1a Causes Increased Oogonia, Reduced Phase III and Phase IV Follicles and Retarded Follicle Growth

To investigate the roles of Tsp1a in follicle development, the gross morphology and histology of the ovaries in the WT, *tsp1a*^−/−^ fish were analyzed. At 60 dah, histological analysis revealed no difference between the *tsp1a*^−/−^ and WT ovaries. The gonadosomatic index (GSI) and the number of germ cells at different phases displayed no significant differences compared with those of the WT fish (Figure 3A–D). At 120 dah, the GSI of the *tsp1a*^−/−^ fish showed no significant differences from that of the WT fish (Figure 3G). Histological observation showed more oogonia and less phase I and II follicles were observed in the *tsp1a*^−/−^ ovaries (Figure 3E,F). Consistently, statistical analysis showed that the number of oogonia and phase II follicles in the *tsp1a*^−/−^ ovaries was significantly higher and lower than that of the WT ovaries, respectively (Figure 3H). At 180 dah, the GSI in the *tsp1a*^−/−^ fish was significantly lower compared with the WT fish (Figure 3K). Morphological analysis showed that the ovaries of the WT fish contained plenty of vitellogenic oocytes while the ovaries of the *tsp1a*^−/−^ fish contained a few vitellogenic oocytes (Figure 3I′,J′). Histological analysis showed more phase I and II follicles and less phase III and IV follicles were observed in the *tsp1a*^−/−^ ovaries (Figure 3I,J). Consistently, statistical analysis showed that the number of oogonia, phase I and II follicles were significantly higher, and the number of phase III and IV follicles was significantly lower in the *tsp1a*^−/−^ ovaries than that of the WT ovaries (Figure 3L). 

### 2.4. Loss of tsp1a Alters Expression Pattern of Genes Related to Ovary Development

To explore the possible reason for increased oogonia, reduced phase III and phase IV follicles and retarded follicle growth, we compared the transcriptomes of WT and *tsp1a*^−/−^ ovaries at 120 dah using RNA-seq. Transcriptomic analysis of the WT and *tsp1a*^−/−^ ovaries indicated significant differences in gene expression (Figure 4A). Kyoto Encyclopedia of Genes and Genomes (KEGG) enrichment analysis of the differentially expressed genes (DEGs) showed that expressions of *tsp* family, DNA replication, cAMP and MAPK signaling pathway genes were significantly changed (Figure 4B). Significant up-regulation of *tsp* family genes, namely *tsp1b*, *tsp2*, *tsp4a*, and *tsp4b,* were observed in the *tsp1a*^−/−^ ovaries (Figure 4C). Genes related to DNA replication, including *pcna*, were significantly up-regulated in the transcriptomes of *tsp1a*^−/−^ ovaries (Figure 4D). However, expressions of cAMP signaling pathway genes were significantly down-regulated, of which *fshβ* was important for folliculogenesis (Figure 4E). Expressions of MAPK signaling pathway genes, *tgfβ2*, *tgfβ3*, *fgfr2*, *fgf6*, *fgf13*, *fgfr3*, *igf2*, *mapk10*, *mapk8ip1*, *map3k13* and *map3k4,* were also significantly down-regulated (Figure 4F). 

### 2.5. Up-Regulation of PCNA Expression in tsp1a^−/−^ Ovaries in Comparison with WT Ovaries 

By IHC, the PCNA (cell proliferation marker) positive signal, which was observed in the oogonia, was increased in the *tsp1a*^−/−^ ovaries compared with that in the WT ovaries at 120 and 180 dah (Figure 5A,B,D,E). Consistently, statistical analysis reveal that significantly increased PCNA positive signal in the *tsp1a*^−/−^ ovaries compared with that of the WT ovaries at 120 and 180 dah (Figure 5C,F). These results indicated that the loss of *tsp1a* resulted in increased oogonia.

### 2.6. Effects of tsp1a Homozygous Mutation on the Expression of Cyp19a1a and Serum E2 Level 

Immunofluorescence analysis showed that the expression of 17β-estradiol (E2) synthase (Cyp19a1a) was decreased in the *tsp1a*^−/−^ ovaries in comparison with the WT ovaries at 180 dah (Figure 6A,B).Consistently, quantified positive signal of Cyp19a1a was significantly lower in the *tsp1a*^−/−^ ovaries than that of the WT ovaries at 180 dah (Figure 6C). Further, EIA analysis showed that the serum E2 was significantly decreased in the *tsp1a*^−/−^ ovaries compared with that of the WT ovaries at 180 dah (Figure 6D). 

## 3. Discussion

In mammals, TSP1 was expressed in a wide range of tissues examined including granulosa cells of the ovaries, lung, skeletal muscle and so on [22,23,24], while the role of TSP1 in the ovaries remains to be elucidated. In teleosts, two copies of *tsp1* (*tsp1a* and *tsp1b*) were found due to the teleosts specific genome duplication [15]. In Nile tilapia, the expression of *tsp1a* was detected in the granulosa cells of the ovaries [15]. But the function of *tsp1a* in teleosts, especially in gonad development, remained to be elucidated. 

The cellular localization of TSP1 is basically the same in ovaries of mammals. In bovine, TSP1 was reported to be expressed in ovarian granulosa cells and theca cells, with high expression in the small follicles and low expression in the pre-ovulatory follicles [13]. In marmoset, TSP1 was expressed in granulosa cells of secondary, tertiary, pre-ovulatory, and atretic follicles. Expression peaked at the tertiary stage, with reduced expression at the pre-ovulatory stage [14]. In rat, TSP1 showed a similar pattern, with expression in ovarian granulosa cells during the early antral and ovulatory phases [5]. In teleosts, the expression localization of *tsp1* mRNA has not been reported except in tilapia. Tsp1 is expressed in the early embryo of zebrafish, but the cellular localization has not been reported [29]. In tilapia, in situ hybridization analysis showed that *tsp1a* was expressed in ovarian granulosa cells [15]. In the present study, Tsp1a was detected in the oogonia, granulosa cells of phase I to phase IV follicles in the ovaries by using the polyclonal antibody of Tsp1a. Expression of *tsp1a* mRNA in the ovaries gradually increased during folliculogenesis, with a remarkably increased level at the vitellogenic stage [15]. Similarly, expression of Tsp1a in the granulosa cells was low in the phase I and II follicles, then increased in phase III and IV follicles. Taken together, Tsp1 was expressed in ovarian granulosa cells in mammals as well as in teleosts. These results indicated that Tsp1 may play an important role in ovarian folliculogenesis of vertebrates.

Roles of TSP1 in the ovaries of vertebrates were poorly understood. Mice lacking *TSP1* was subfertile and exhibited ovarian hyper-vascularization and altered ovarian morphology [21]. In the present study, homozygous mutation of *tsp1a* resulted in increased oogonia, decreased phase III and IV follicles and retarded follicle growth in tilapia. To be sure the defects of the mutants were not caused by abnormal angiogenesis, we performed IF using antibody against Vegfr2 (vascular endothelial marker) with ovaries sections at 120 dah. No differences in angiogenesis were observed between the *tsp1a* mutants and the WT fish (Appendix A), which was probably due to the compensation of *tsp* family genes that were significantly up-regulated in the *tsp1a* mutant fish. 

In *tsp1a* homozygous mutants, a large number of oogonia were observed and the number of PCNA positive cells was significantly higher compared with that of the WT fish. Consistently, transcriptomic analysis revealed that expressions of DNA replication-related genes including *pcna* were significantly up-regulated in *tsp1a* homozygous mutants. These results indicated that lacking *tsp1a* caused up-regulation of DNA replication related genes, and then resulted in increased oogonia. In the ovaries, cAMP is of utmost importance as a second messenger for growth and development of follicles in all stages [30,31,32]. Studies revealed that cAMP response element-binding protein 1 (creb1) and the cAMP analogue, dbcAMP were involved in the down-regulation of porcine and rabbit ovarian cell proliferation, respectively [33,34]. In addition, cAMP inhibited rat hepatocellular carcinoma cell proliferation and enhanced cell differentiation [35]. In the present study, expressions of cAMP and MAPK signaling pathway genes were significantly down-regulated in *tsp1a* mutant fish. In addition, down-regulation of Cyp19a1a expression and serum E2 level in the *tsp1a* mutant fish in the ovaries were observed. These results indicated that lacking *tsp1a* resulted in increased proliferation and inhibited differentiation of oogonia, which in turn, resulted in increased oogonia, reduced secondary growth follicles and decreased E2. But at the vitellogenic stage, the defects were recovered as most of the oogonia differentiated into the phase III and phase IV stage, which was probably due to the compensation of *tsp* family genes that were significantly up-regulated in the *tsp1a* mutant fish. 

In summary, using CRISPR/Cas9-mediated gene knockout technology, we generated the *tsp1a* homozygous mutant fish, which providing genetic evidence for the role of *tsp1a* in the process of folliculogenesis in teleosts. Homozygous mutation of *tsp1a* resulted in increased oogonia, decreased phase III and IV follicles and retarded follicle growth. Taken together, our results suggested that *tsp1a* was important for folliculogenesis in female tilapia.

## 4. Materials and Methods

### 4.1. Animals

Nile tilapia (*Oreochromis niloticus*) was reared in recirculating aerated freshwater tanks at 26 °C under a natural photoperiod. All-XX progenies were obtained by crossing a pseudomale (XX male, producing sperm after hormonal sex reversal) with a normal XX female. All-XY progenies were obtained by crossing an YY supermale with an XX female. Animal experiments were performed following the regulations of the Guide for Care and approved by the Institutional Animal Care and Use Committee of Southwest University (No. IACUC-20181015-12, 15 October 2018). 

### 4.2. Establishment of tsp1a Homozygous Mutants by CRISPR/Cas9

CRISPR/Cas9 was performed to knockout *tsp1a* in tilapia as described previously [27]. Briefly, the guide RNA and Cas9 mRNA were co-injected into one-cell-stage embryos at a concentration of 500 and 1000 ng/μL, respectively. Twenty injected embryos were collected 72 h after injection. Genomic DNA was extracted from pooled control and injected embryos and used to access the mutations. DNA fragments spanning the target site was amplified. The mutated sequences were analyzed by restriction enzyme digestion with HpyAV and Sanger sequencing [36]. 

Heterozygous F1 offspring were obtained by F0 XY male founders mated with WT XX females. The F1 fish were genotyped by fin clip assay and the individuals with frame-shift mutations were selected. XY male and XX female siblings of F1 generation, carrying the same mutation, were mated to generate homozygous F2 mutants. The *tsp1a* F2 mutants were screened using restriction enzyme digestion and Sanger sequencing. The genetic sex of each fish was determined by genotyping using sex-linked marker (marker 5) as described previously [25].

### 4.3. Sampling and Histological Analysis

For each fish sampled for histological analysis, the somatic growth parameters including body weight and gonad weight were measured before processing. After sampling and morphometric measurement [37], gonads of the WT and *tsp1a*^−/−^ fish were dissected at 60, 120 and 180 dah. Then the gonads were fixed in Bouin’s solution for 24 h at room temperature, dehydrated and embedded in paraffin. Tissue blocks were sectioned at 5 μm thickness. The sections were deparaffinized, hydrated, stained with hematoxylin and eosin (H and E), or used for immunofluorescence (IF) and immunohistochemistry (IHC) analysis. 

### 4.4. Immunofluorescence (IF) and Immunohistochemistry (IHC)

In the present study, Tsp1a polyclonal antibody (peptide antigen: RDDNSVYDLFELVKVPSKNHGVTLVKGDDPYSPAYKILNPDLIPPVPENSFRDLIDSIHAERGFLLLLNFKQFRRTRGSLLTVEKKDGSGPVFEIVSNGKANTLDIVFSTENKQQVVSIEDVGLATGQWKNITLFVQEDWAKLYVGCDEVNTAELDAPIQSILTQETPASAQLRVGKGAVKDKFTGVLQNVRFVFGTTLEAILRNKGCQSAMTDTMVLRNLNGSSAIRTEYSGHKTKDLQMVCGFSCEDLLSMFKELKGLGVVVKELSTELRKLTDENKLIKSRIGIHSGVCIHNGIVRKNRDEWTVDDCTECTCQNSATVCRKISCPLIPCANATVPDGE) was produced (Abiotech, Jinan, China) to characterize the cellular localization in tilapia ovaries. Tsp1a antibody was diluted at 1:500 for use. Vegfr2 (ProteinTech, Wuhan, China) antibody was used to detect blood vessel density of the ovaries. PCNA antibody (Cusabio, Wuhan, China) was used to detect cell proliferation. Cyp19a1a antibody was a gift from Professor Yoshitaka Nagahama, National Institute for Basic Biology, Okazaki, Japan [38]. Antibodies against Vrgfr2, PCNA and Cyp19a1a were diluted at 1:500, 1:500 and 1:2000, respectively. The specificity of these antibodies was checked previously [39,40]. For IF, Alexa Fluor 488- 586- and 594-conjugated secondary antibodies (Invitrogen, Shanghai, China) were diluted 1:500 in blocking solution and incubated with tissue to detect the primary antibodies. The nuclei were stained by 4′,6′-diamidine-2-phenylindole-dihydrochloride (DAPI) (Invitrogen, Carlsbad, USA). IHC analyses were performed as described previously [41]. For IHC, the second antibody (HRP-conjugated goat anti-rabbit IgG, 1:1000 dilution) was used to detect the primary antibody. Diaminobenzidine tetrachloride (DAB) was applied for the color reaction. Slides were first counterstained with hematoxylin, and then dehydrated and mounted [42]. Photographs were taken under an Olympus BX51 light microscope (Olympus, Tokyo, Japan). Finally, the positive signals were quantified using image J software (National Institutes of Health, Bethesda, MD, USA) according to the instructions. 

### 4.5. Transcriptome Sequencing and Analysis

Total RNA was extracted from *tsp1a*^−/−^ and WT XX ovaries of fish at 120 dah using Trizol reagent (Invitrogen, Carlsbad, CA, USA) according to the manufacturer’s instructions. The extracted RNA was further treated with DNaseI (RNase-free, 5 U/μL) to eliminate genomic DNA contamination. Poly (A) + mRNA was purified using the DynaBeads mRNA purification kit (Life Technologies, Carlsbad, CA, USA). Six cDNA libraries, three for *tsp1a*^−/−^ and three for WT and each with RNA for three ovaries, were constructed. Sequence reads (2*100 bp pair-end sequencing) pools were generated on an Illumina HiSeq™ 2000 instrument from the Beijing Genomics Institute at Shenzhen (BGI Shenzhen, China) from the cDNA libraries. The clean reads from the transcriptomes were obtained from raw data by filtering out adaptor-only reads and low-quality reads (reads with *q* value ≤ 20). The reference genome and gene data from *Oreochromis niloticus* (Orenil1.0) were downloaded from the Ensembl web site (http://www.ensembl.org/Oreochromis_niloticus/Info/Index). Clean reads from each library were aligned to the reference genome using TopHat v2.0.6. This program allows multiple alignments per read (up to 20 by default) and a maximum of 2 mismatches when mapping the reads to the reference genome. The fragments per kilobase of exon per million fragments mapped (FPKM) method was used to calculate gene expression levels. The differentially expressed genes (DEGs) were classified according to the following criteria: genes meeting both “*q* value < 0.05” and “log2 (*tsp1a* KO_FPKM/WT_FPKM) > 1” statistical criteria were classified as up-regulated genes; in contrast, genes meeting both “*q* value < 0.05” and “log2 (*tsp1a* KO _FPKM/WT_FPKM) < −1” were classified as down-regulated genes. The Gene Ontology (GO) of the DEGs was analyzed by WEGO online tool. To study the biological pathways of the identified up/down-regulated genes, we mapped these DEGs to pathways in the Kyoto Encyclopedia of Genes and Genomes (KEGG) using the KOBAS web server (https://kobas.cbi.pku.edu.cn/)

### 4.6. Follicle Counting

Fixed ovaries were embedded in paraffin and sectioned at 5 μm thickness. The ovaries of 6 fish of each genotype (WT and *tsp1a*^−/−^ XX fish) were dissected at 60, 120 and 180 dah. Follicles from the median sections of ovaries (*n* = 5) were counted for statistical analysis. The histological classification of the oocytes was performed as described previously [37,43]. 

### 4.7. Measurement of Serum E2 Level

Blood samples were collected from 6 fish of each genotype (WT, *tsp1a*^−/−^ fish) at 180 dah and kept at 4 °C overnight. The serum (*n* = 5) was collected after centrifugation and stored at −20 °C until use. Serum E2 levels were measured using the Enzyme Immunoassay Kit (Cayman, MI, USA). Sample purification and assays were performed according to the manufacturer’s instructions.

### 4.8. Quantitative Reverse Transcription Polymerase Chain Reaction (qRT-PCR)

Ovaries of the WT and *tsp1a*^−/−^ XX fish were dissected at 5, 30, 60, 90 and 180 dah for gene expression assay. Total RNA (1.0 μg) was extracted and reverse transcribed using PrimeScript RT Master Mix Perfect Real Time Kit according to the manufacturer’s instructions (Takara, Dalian, China). qRT-PCR was performed on an ABI7500 qRT-PCR machine (Thermo Fisher, Waltham, MA, USA), according to the protocol of SYBR1 Premix Ex TaqTM II (Takara, Dalian, China). The relative abundance of key genes in the ovaries was evaluated using the formula R = 2^−ΔΔ*C*t^. The reference gene tilapia *eef1a1a* was used to normalize the expression values. 

### 4.9. Western Blot

Ovaries from the WT and homozygous *tsp1a* mutants (*n* = 5/genotype) at 180 dah were dissected and pooled. Total proteins were extracted and diluted to a final concentration of 20 mg/ml. Western blot was carried out as described previously [41]. Antibody against Tsp1a was diluted at 1:500. The abundance of α-tubulin was examined as a loading control using rabbit anti-α-tubulin (Proteintech, Wuhan, China) at 1:1000. Horseradish peroxidase-conjugated goat anti-rabbit antibody (Beyotime, Shanghai, China) was used as the secondary antibody at 1:1000. Immunofluorescence signal was detected with BeyoECLPlus Kit (Beyotime, Shanghai, China) and was visualized on a Fusion FX7 (Vilber Lourmat, East Sussex, France).

### 4.10. Statistical Analysis

All data were presented as the mean ± standard deviation (mean ± SD). Two-tailed unpaired Student’s *t*-test was used to determine statistically significant difference between two groups. One-way ANOVA was performed for comparisons with more than two groups followed by Tukey’s test. Statistics analyses were performed using GraphPad Prism 8 software package (GraphPad Software, La Jolla, CA, USA). In all analyses, *p* < 0.05 was considered to be significantly different.

## Figures and Tables

**Figure 1 ijms-21-05893-f001:**
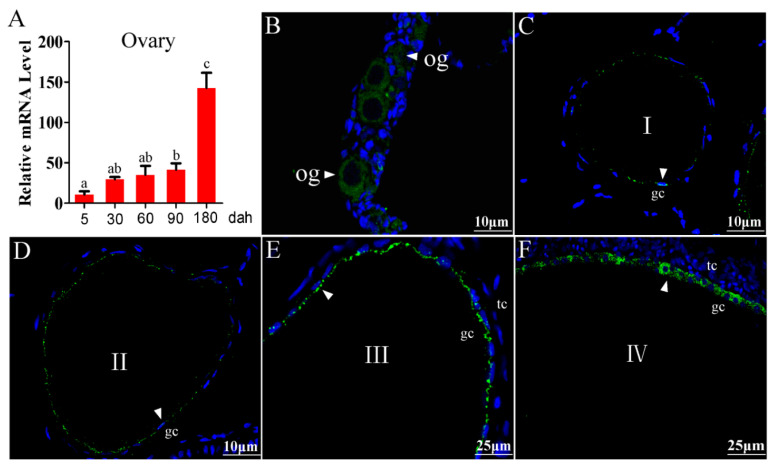
Immunofluorescence (IF) localization of Tsp1a in tilapia ovaries. (**A**) Expression of *tsp1a* mRNA in the tilapia ovaries at different developmental stages, as determined by qRT-PCR. Data are expressed as the mean ± SD of three different gonadal pools at each developmental stage. Different letters above the error bar indicate statistical differences at *p* < 0.05 as determined by one-way ANOVA followed by Tukey test. (**B**–**F**) Cellular localization of the Tsp1a protein in tilapia ovaries, as determined by IF. og, oogonia; I to IV, phase I to IV follicles; gc, granulosa cell; tc, theca cell. Green fluorescence represents the Tsp1a signal (white arrow). Blue fluorescence represents the DAPI signal. Tsp1a was expressed in the oogonia and granulosa cells of phase I to phase IV follicles.

**Figure 2 ijms-21-05893-f002:**
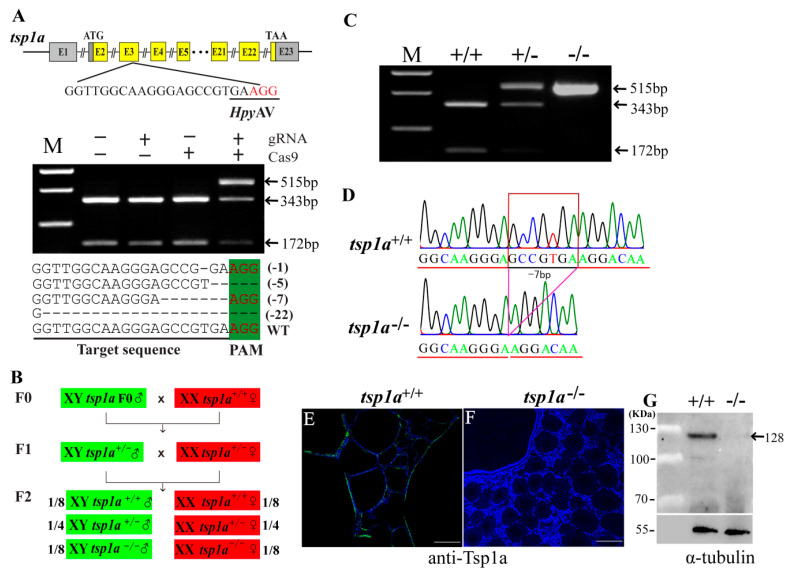
Efficient mutation of *tsp1a* by CRISPR/Cas9. (**A**) Gene structures of *tsp1a* showing the target site and the HpyAV restriction site. The Cas9 mRNA and gRNA were added as indicated. Sanger sequencing results from the uncleaved bands were listed. The PAM (protospacer adjacent motif) is marked in green. Deletions are marked by dashes (–) and numbers to the right of the sequences indicate the loss of bases for each allele in parentheses. The mutant fish which contains 7 base-pairs deletion was used for homozygous mutant construction of *tsp1a*. (**B**) Schematic diagram showing the breeding plans of *tsp1a* F0 to F2 fish. (**C**) Identification of F2 genotypes by restriction enzyme digestion assay. (**D**) Sequencing results of *tsp1a* gene from WT and homozygous mutant fish. (**E**,**F**) Immunofluorescence was used to test the specificity of the Tsp1a antibody. Cells with a green color indicated the positive signal. Blue fluorescence represents the DAPI signal. (**G**) Validation of the specificity of Tsp1a antibody by western blot. The Tsp1a protein was present in WT tilapia, whereas no band was observed in the mutants. Specific bands were indicated by arrow.

**Figure 3 ijms-21-05893-f003:**
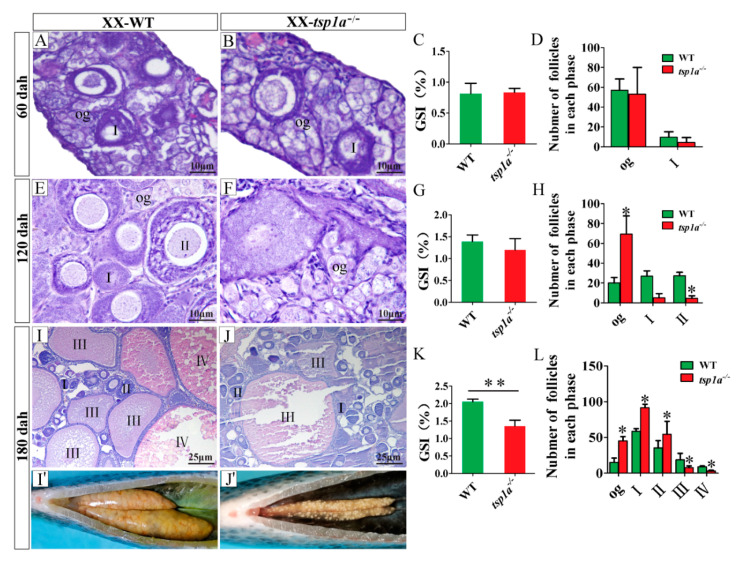
Morphological and histological analyses of WT and *tsp1a*^−/−^ ovaries at 60, 120 and 180 dah. (**A**,**B**,**E**,**F**,**I**–**J’**) Morphological and histological observation. (**C**,**G**,**K**) Gonadosomatic index (GSI) (*n* = 5). (**D**,**H**,**L**) Statistical analysis of oocyte counting (*n* = 5). Differences between the mutants and WT were tested by two-tailed unpaired Student’s *t*-test, * *p* < 0.05; ** *p* < 0.01. Results were presented as the mean ± SD in C-D, G-H, and K-L. dah, days after hatching. WT, wild type. Og, oogonia. I to IV, phase I to phase IV follicles.

**Figure 4 ijms-21-05893-f004:**
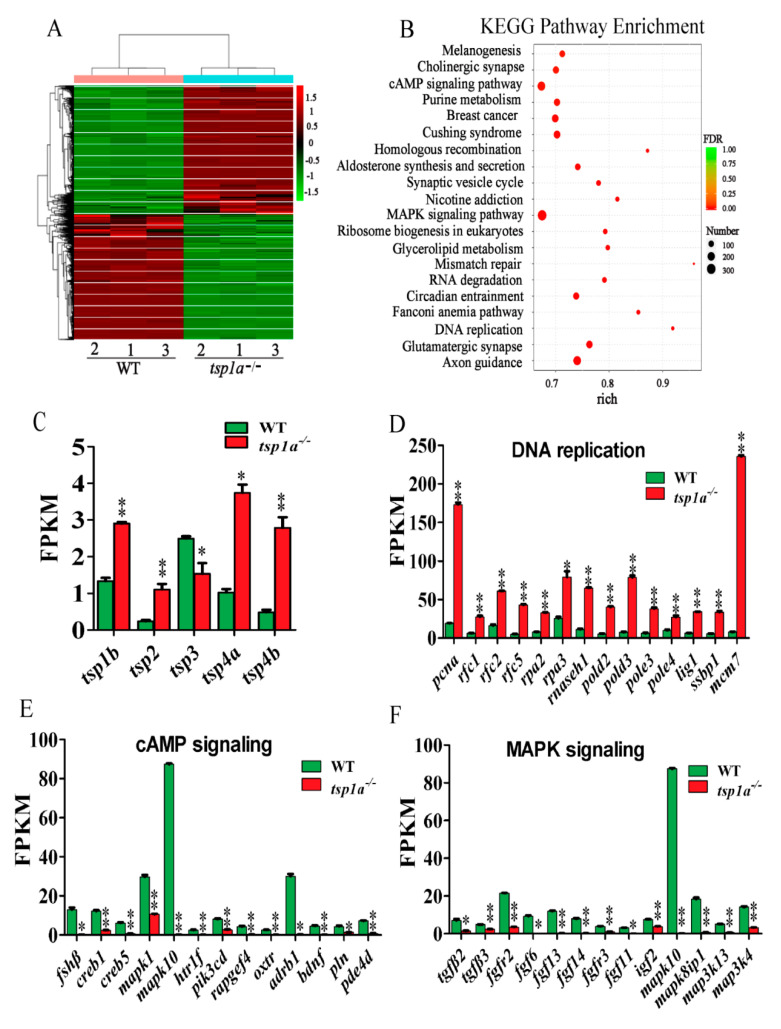
Comparative transcriptomic analysis of genes and molecular pathways dysregulated in *tsp1a*^−/−^ ovaries. Six cDNA libraries, three for *tsp1a*^−/−^ and three for WT and each with RNA for three ovaries, were constructed and sequenced. (**A**) Heat map of the expression of genes (FPKM, fragments per kilobase of exon per million fragments mapped) in *tsp1a*^−/−^ and WT ovaries. (**B**) KEGG enrichment analysis of DEGs. Scatterplot of enriched KEGG pathways for DEGs screened from *tsp1a*^−/−^ vs. WT fish. The enrichment factor indicates the ratio of the DEGs number to the total gene number in a certain pathway. The size and color of the dots represent the gene number and the range of false discovery rate (FDR), respectively. (**C**–**F**) Comparative transcriptomic analysis between *tsp1a*^−/−^ and WT ovaries reveals a significant up-regulation of DNA replication-related genes and *tsp* family genes and significant down-regulation of cAMP and MAPK signaling pathway genes. Data were expressed as the mean ± SD of triplicates. Differences between groups were statistically examined with two-tailed unpaired Student’s *t*-test; * *p* < 0.05, ** *p* < 0.01.

**Figure 5 ijms-21-05893-f005:**
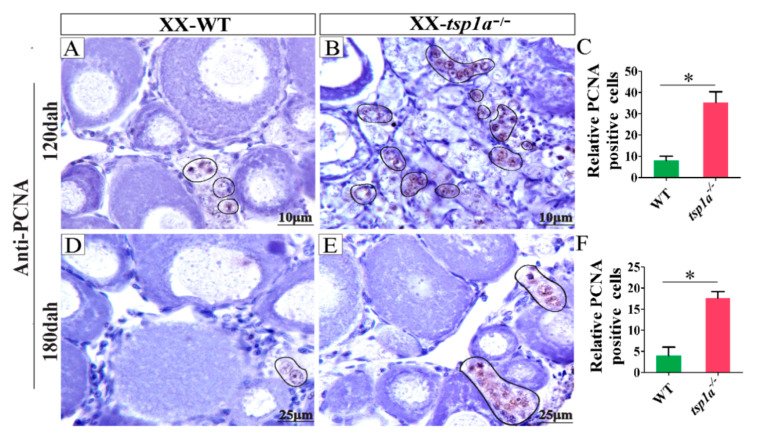
Up-regulation of PCNA expression in *tsp1a*^−/−^ ovaries in comparison with WT ovaries at 120 and 180 dah. (**A**,**B**,**D**,**E**) Immunohistochemistry analysis. (**C**,**F**) The statistical analysis of the positive signals (*n* = 5, and five sections for per sample were counted). The PCNA positive germ cells were marked by circles. Differences between the mutants and WT were tested by two-tailed unpaired Student’s *t*-test, * *p* < 0.05. Results were presented as the mean ± SD in (**C**) and (**F**).

**Figure 6 ijms-21-05893-f006:**
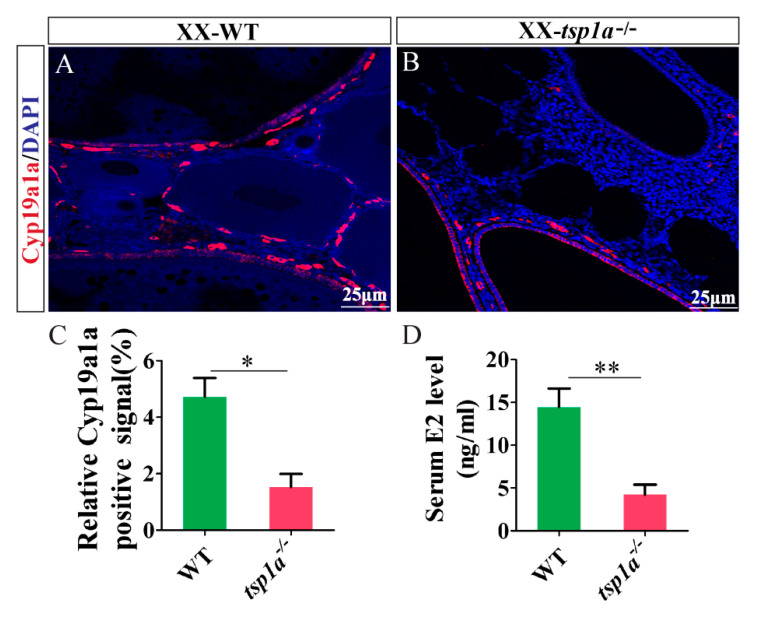
Down-regulation of Cyp19a1a expression and serum E2 level in *tsp1a*^−/−^ ovaries in comparison with WT ovaries at 180 dah. (**A**,**B**) Immunofluorescence analysis. Red fluorescence represents the Cyp19a1a signal. Blue fluorescence represents the DAPI signal. (**C**) The statistical analysis of the positive signals (*n* = 5, and five sections for per sample were counted). (**D**) Serum E2 EIA analysis (*n* = 5). Differences between the mutants and WT were tested by two-tailed unpaired Student’s *t*-test, * *p* < 0.05; ** *p* < 0.01. Results were presented as the mean ± SD in C and D.

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
