# Peer review of "Regulation of Female Folliculogenesis by Tsp1a in Nile Tilapia (Oreochromis niloticus)"

_ijms, 2020, doi:10.3390/ijms21165893_

Round 1

Reviewer 1 Report

In the manuscript “Regulation of Female Folliculogenesis by Tsp1a in Nile Tilapia (Oreochromis niloticus)” authors discussed tsp1a essential role in ovarian folliculogenesis. The concept is interesting also for the possible clinical implications in humans. The manuscript should be improved in English language and specific details included to avoid misunderstandings. The authors should eventually consider further investigations on a larger number of specimens to validate RNA-seq data (by quantitative real-time RT-PCR) and demonstrate the biological meaning of DE genes.  

Specific Comments

  • Different fonts were used in the text. Check.
  • English language should be revised (e.g. line 57: In mice, lacking TSP1 was subfertile; line 229: were significantly down-regulation down-regulated)
  • Specify the sample number used (e.g. line 324).
  • In results and discussion, the presentation of the data should be improved.
  • Some inaccuracy should be amended in the text (e.g. line 143: Figure 6A->4A).
  • In results (line 147) the authors say to detect 34 DEGs including pcna related to DNA replication, of which 2 were up-regulated and 2 were down-regulated in the tsp1a-/- ovary transcriptomes (Figure 4D). In this sentence, the numbers should be eventually revised.
  • Figure 1A. Include the statistical significance values at which the letters above error bars refer to.
  • Figure 4A. Specify the type of expression data represented.  
  • Showing the final number of samples analysed in the legend of each figure could be more informative.

Reviewer 2 Report

In this manuscript the authors investigate the role of Tsp1a in the Tilapia gonad. Tsp1 has previously been shown to be involved in controlling angiogenesis in the mature mammalian ovary in knockdown and knockout mouse models. The authors show the expression of Tsp1a in the developing gonad, and use immunofluorescence to localise this expression to the granulosa cells of the follicles. They use CRISPR/Cas9 to first validate cutting of the locus and then generate G1 animals containing a 7bp deletion in exon 3 of the locus. These animals were bred to generate knockout G2 animals. The mutant animals developed normally but the oocytes showed less and delayed maturation. A RNAseq analysis was carried out and genes involved in folliculogenesis were identified and many of these were confirmed by QPCR.

PCNA was upregulated in small ‘follicles’ and the lack of mature follicles led to a reduction in aromatase and lower oestrogen (E2) levels. This is a well presented and interesting study.

Major comment:

Previous work showed an increase in angiogenesis in mutant mouse gonads. The authors should stain the gonads with an endothelial marker to show this phenotype in the mutant animals. They state ‘However, abnormal angiogenesis was not observed in tsp1a mutant tilapia,’ yet this data is not shown.

Minor points

  1. What is the abbreviation dah?
  2. The sequence traces are too clean. What programme was used? These do not look like the true output files.
  3. A western blot on the novel antibody is needed to show specificity.
  4. Point out that aromatase is cyp19a1.
